# Concept of Evaluation of Mineral Additives’ Effect on Cement Pastes’ Durability and Environmental Suitability

**DOI:** 10.3390/ma14061448

**Published:** 2021-03-16

**Authors:** Robert Figmig, Adriana Estokova, Miloslav Luptak

**Affiliations:** 1Faculty of Civil Engineering, Institute of Environmental Engineering, Technical University of Kosice, 04200 Kosice, Slovakia; adriana.estokova@tuke.sk; 2Faculty of Materials, Metallurgy and Recycling, Institute of Materials and Quality Engineering, Technical University of Kosice, 04200 Kosice, Slovakia; miloslav.luptak@tuke.sk

**Keywords:** cement composite, silica fume, zeolite, w/b ratio, cement paste porosity, cement matrix permeability

## Abstract

This experimental study focuses on the assessment of mineral additives and their incorporation into cement composites (CC). The assessment was based on a holistic approach to the performance of the durability properties of CC. Environmental suitability was also taken into consideration. In the experiments, cement pastes with w/c ratios of 0.3, 0.4, and 0.5, respectively, were prepared. Natural zeolite (NZ) and densified silica fume (SF) at doses of 7.5 and 15.0 wt.% of cement were used as the investigated (replacement) materials. Their effects (including development over time) on density, strength (flexural and compressive), porosity by water absorption, permeability by rapid chloride penetration (RCP) test, phase content by thermal analysis, and hydration progression, were observed. The results were then used to propose an evaluation approach. Natural zeolite was used for its known pozzolanic activity and classification as a supplementary cementitious material (SCM). In contrast SF acted as a filler in cement pastes, and thus did not have a direct positive effect on durability. The concept of comprehensive analysis for unknown additive classification is proposed to expressly differentiate between SCM, inert, and improving mineral additive. This concept could be applied to the assessment of mineral additives with regards to the durability and suitability of cement composites.

## 1. Introduction

The durability of building materials can be considered in several respects, but it is currently clearly understood as one of the pillars of sustainability in the construction sector. Sustainability, another aspect penetrating industrial production, is an integral part of intergenerational equity relating to the right to a beneficial environment—one of the common values of humankind [1]. The sustainability of concrete can be succinctly described from three points of view:


(a)technical—representing material functionality,(b)environmental—linked to saving of natural resources through prolongation of the construction lifetime, and(c)economic—frugality parameter. All these aspects are mutually interconnected in sustainability holistic approach [2,3].


Durability is a priority factor for concrete, considering its purpose as a structural material. It consists of performance (preservation of properties) and time (long-term lifetime) aspects [4,5]. It is determined by stipulation specification and consequently by fresh concrete composition, as well as by production technology, processing, and curing [6,7,8]. The concept of several scales of concrete [9,10,11] related to durability is given in Figure 1a. It is well-known that the durability of concrete mainly relates to porosity. Even compressive strength as a basic CC (cement composites) durability property is well described as dependence on porosity that relates to the w/c ratio [12,13,14]. Besides plastic CC properties (rheology, density, and setting time), there are various tests for the hardened state of CC based on mechanical, physical, thermophysical, chemical, or physico-chemical properties [15]. The various alternatives of system porosity-permeability [16] with effects on key parameters [12] are given in Figure 1b.

According to Figure 1b, four basic cases of open pore system can occur within a cement paste microstructure (inaccessible pores will not be considered) [17,18]:


High porosity and high permeability (up left)—increases water absorption, intensifies liquid migration, and has a negative effect on strength and deformation.High porosity and low permeability (up right)—negatively affects water absorption and strength and deformation, but leads to lower liquid migration.Low porosity and high permeability (down left)—decreases water absorption and increases liquid migration.Low porosity and low permeability (down right)—this is the ideal case, wherein the lowest water absorption and liquid migration are achieved as well as the highest strength and lowest deformations.


The crucial aspects for CC durability are, therefore, the quantity and quality of cement paste. Quantity affects overall durability due to its much lesser durability in comparison with the aggregate. With increase in cement paste amount of same w/c, the volume of the pores increases, thus increasing porosity and permeability [19]. The main factor of cement paste quantity is the aggregate (mainly its granularity and grain shape) [20,21,22,23,24,25,26,27]. This is due to voids that are created between grains that have to be fulfilled by cement paste. Moreover, appropriate workability have to be ensured. Currently, aggregate packing (among particle packing models) are in focus [28,29,30]. Cement paste (CP) quality is the result of a porous system (porosity, capillarity). It has impacts on all mechanical, physical, and chemical properties, while the most important parameter is the water-to-cement (binder) ratio w/c (w/b) [31,32]. The temperature of input materials is an often neglected aspect, even though increasing temperature negatively influences the workability of plastic concrete. This consequently leads to the additional need for water [33,34]. Increase in cement paste quality can be done by several ways: (1) decreasing w/c ratio—however, this increases the amount of cement paste and, thus, could be counterproductive [35], (2) using latent hydraulic (GGBFS) or pozzolanic materials (fly ash, diatomite, silica fume, zeolite, metakaolin, etc.) [36], and (3) active air—considering entraining air as beneficial for general durability is controversial [37]; however, in our opinion, only an environment with freeze-thaw cycles is beneficial for hardened CC, since using an air-entraining admixture increases permeability and, thus, overall porosity [38,39,40,41].

Current research on concrete durability is focused on adjusting its durability, while minimizing the content of its constituents; for example, ordinary Portland cement (OPC) with a high carbon footprint, since production of OPC is connected to high heat energy demand [42,43,44]. This is based on using Supplementary Cementitious Materials (SCM), while durability properties are, at least, preserved [45,46,47]. Beushausen et al. [35] proposed a performance-based approach for concrete durability in the designing stage as an alternative to the traditional prescriptive approach. In [48], Juhart et al. presented a new method based on a systematic design approach for the efficient use of ordinary Portland cement (OPC) in cement-based materials, in combination with inert mineral fillers. They successfully applied it to identify and characterize proper fillers and to find optimum mix-ratios for eco-efficient pastes consisting of OPC, very fine micro-fillers, and fine eco-fillers.

Tran et al. [49] concluded that using natural zeolites in the cement composites on the one hand reduces compressive strength and workability; their addition can considerably enhance other engineering properties, such as shrinkage, chloride penetration, water permeability, carbonation resistance, and sulphate resistance. Shahmansouri et al. [50] studied eco-friendly geopolymer concrete incorporating silica fume and natural zeolite, while ascertaining a worsening effect of additives on workability, but a beneficial compressive strength (silica fume) and long-term compressive strength (natural zeolite). In [51], Azad et al. studied the effect of zeolite on the environmental and physical characteristics of green concrete filters, which they declared was enhancing the filter ability in reducing pollution parameters of wastewater, as well as the no-deteriorative effect of using zeolite on physical properties of porous concrete. Nas and Kurbetci [52] showed improved durability against freeze-thaw cycles as well as against penetration of chloride ions using NZ (natural zeolite). Ahmad [53] studied different ratios of silica fume replacement on compressive strength and hardness, also considering the cost, while he presented that 15% of Portland cement replacement is most beneficial. Sasanipour et al. [54], in their study on the effect of silica fume on durability of self-compacting concrete made with waste recycled concrete aggregates, concluded that SF (silica fume) plays an important role in improving durability performance, since it decreases water absorption and significantly increases electrical resistivity and chloride ion penetration resistance. Khodabakhshian et al. [55] performed an investigation on the durability performance of structural concrete containing silica fume, concluding on some negative effects of SF on concrete rheology, but showing a positive effect on compressive strength, even in corrosive environments (sodium and magnesium sulphate, sulphuric acid); SF also slightly improved water absorption. In our previous study [56], we found that increasing the amount of zeolite supplement leads to a decrease in workability, while studying its utilization in the precast industry. Berenguer et al. [57] studied the durability of concrete structures with sugar bagasse ash, while considering it (sample A) as a pozzolanic material to improve the durability properties of concrete. In [58], Pan et al. investigated the effect of the rheological properties of fresh concrete on shotcrete-rebound based on different additive components.

This paper focuses on examining select known mineral additives as materials adjusting the binder phase of CC in relation to design and production durability, thus resulting in sustainable concretes. This study is based on experiments performed exclusively on cement pastes with various w/b ratios and cement replacements. In this manner, the impact of aggregate (filler phase) on the results could be eliminated and, thus, a higher amount of primary materials (cement, additives) in unit volume than in mortars or concretes could be tested. In addition, gravitation pore, ITZ (interfacial transition zone) zone pores, and partially entrapped voids creation could be neglected. This approach is more suitable in investigating cement composite durability and sustainability. Moreover, thermal analysis was applied to study the hydration processes. The novelty of the paper lies in the proposed method of evaluation and classification of the mineral additives, as per their effects on CC properties relating mainly to durability. The proposed method is based on comprehensive performance analysis, thus eliminating isolated evaluation of the absolute values of the test results. In this article, the filler indicates the low reactivity component in the role of the aggregate. The binder phase covers cement paste as a compound of cement and water with or without active mineral additives.

## 2. Materials and Methods

### 2.1. Input Materials

As reference material, cement CEM II/A-S 42.R (CRH Plant, Turna nad Bodvou, Slovakia) was used. This type of cement is considered to be universal for ready-mix concretes as well as for precast concretes. As SCMs materials, natural zeolite ZeoBau 50 (Zeocem plant, Bystre, Slovakia) and densified silica fume Mapeplast SF (Mapei, Ivanka pri Dunaji, Slovakia) were applied. The chemical composition and specific gravity of materials are given in Table 1. Natural zeolite (NZ) and granulated silica fume (SF) were selected as effective SCMs regarding the durability of cement composites. NZ is a natural mineral with pozzolanic properties, and its carbon footprint is much lower than that of ordinary Portland cement, since NZ does not undergo thermal procession. SF is secondary material produced as a by-product in the production of ferosilicium. Despite its high price, SF is often used when very specific properties (mainly connected with high strength or durability against high corrosive environments) are required [53].

To mix the cement pastes, tap water according to EN 1008:2002-07 [59] was used. To adjust CP mixtures’ workability, polycarboxylate-ether based high-range water reducing (HWR) superplasticizer SF40 (MAPEI, Slovakia) with 34 wt.% of solid content was added as a high-range water reducing admixture.

### 2.2. Mix Design and Labelling

Overall, 15 cement paste batches were mixed. Reference cement pastes and cement pastes with SCMs at two supplement ratios (7.5 wt.% and 15.0 wt.%) were tested at three different w/b ratios (0.3, 0.4, and 0.5, respectively). Mix proportion (binder, water) was the same within each of the binder-to-water ratio. Labelling is given in Table 2. The amount of materials given in Table 2 corresponds to approximately 3.5 L of an individual batch needed to prepare all the samples for the experiment with sufficient excess.

### 2.3. Fresh Cement Pastes and Sample Preparation

The materials were mixed for 120 s in a mixer. To achieve appropriate workability, all the pastes (excluding those with w/b = 0.5) were adjusted by a superplasticizer. Five minutes after first contact of the binder and water (zero time), the flow on Haegermann table with 25 strikes was measured according to EN 1015-3:1999-02 [60]. Then, 30 and 60 min from zero time, the flow dimensions were measured again. Before the test, a batch was remixed for 30 s. After 60 min from the start and finish flow test, the paste was input into molds in 1 layer, vibrated for 5 s, and covered by impermeable foil to prevent the specimen from drying out. At the same time, the density of the fresh pastes (D-mix) was determined. Beam specimens of dimensions 40 × 40 × 160 mm according to EN 196-1:2016-04 [61] were prepared for the strength and water absorption tests; disc specimens were of diameter 42 mm and width 27 mm for the RCPT (rapid chloride penetration test) test.

### 2.4. Water Absorption

The water absorption test was performed using a sample with the same shape as for the flexural strength tests according to STN 73 1316:1989-05 [62]. The tests were conducted at 28 days (WA-28) and 180 days (WA-180) of curing under water. Three samples from every batch and each of two curing times were tested. After pulling the samples out from the water bath, they were surface-dried with a cloth and weighed in saturated and surface dried (ssd) state. They were then dried in an oven at temperature of 105 ± 5 °C until constant mass (Equation (1)) was achieved (oven-dried state “d”)—a minimum of 6 days. Afterwards, WA was calculated according to Equation (2) and the samples set aside for the strength test.
(1)$$m_{d}\equiv \;m_{i}\leftarrow \frac{m_{i-24h}-m_{i}}{m_{i-24h}}\cdot 100<0.1;\;i\;\in \;\left\{24;\;48;\;72;\;\ldots \right\}hours$$
(2)$$WA=\frac{m_{ssd}-m_{d}}{m_{d}}\cdot 100$$
where,*m_d_*—oven dried constant mass of sample rounded to 0.1 (g);*m_ssd_*—mass of saturated sample rounded to 0.1 (g);*WA*—water absorption (wt.%).

### 2.5. Compressive and Flexural Strength

Compressive and flexural strength was investigated after 28 and 180 days of curing under water in saturated (ssd) as well as in oven-dried state (d). For each stated option, three samples for flexural strength and six samples for compressive strength were tested. The contact area of each specimen was brushed to achieve a flat surface and thus precise results. Flexural strength measurement (f_flex_-28-ssd, f_flex_-180-ssd, f_flex_-28-d, f_flex_-180-d) was performed on beams (width 40 mm, height 40 mm, and length 160 mm) with 1 loading in the center (3-point bending set-up) by loading speed of 0.05 MPa/s (approximately equaling 0.02 kN/s). After measuring the dimensions of the fracture area, f_flex_ was calculated according to EN 12390-5:2019-8 [63]. Fracture curves as well as fracture area were optically evaluated and recorded by a camera with macro lens and by the optical microscope AmScope M162C-2L-PB10.

Subsequently, compressive strength (marking similar to flexural strength: f_c_-28-ssd, f_c_-180-ssd, f_c_-28-ssd and f_c_-180-ssd) was executed on six beams’ fractions with loading speed of 0,5 MPa (what equals 0.8 kN/s) using a spacing device and calculated according to EN 12390-3:2019-07 [64].

### 2.6. Rapid Chloride Penetration Test

The rapid chloride penetration test (RCPT) was performed on a cement paste disc with diameter Φ = 42 mm and width a = 27 mm (exposed area A = 5542 mm^2^), as per ASTM C1202-19 [65]. High current flow through all the samples at 28 days of curing caused a rapid increase in the temperature of the sample and a consequent current rise; the measurements were taken at 180 days of curing. The current-diffusivity-temperature relationship is one of the disadvantages of this test method [66,67]. Another drawback is ambiguity—identifying which ions are the carrier of charge—since it could be Cl^-^ from the electrolyte solution as well as ions from the pore solution of the test specimen [15,68,69]. Despite these facts, we consider RCPT (or called the rapid “ions” penetration test) convincing for cement paste comparative analysis. In spite of our attempts to decrease voltage to 20 V, 30 V, 40 V, or 50 V, respectively, when the current decrease was linear (but stayed approximately constant), as well as temperature, we stayed consistent with the standard RCPT procedure—it consisted of applying 60 V voltage to the sample, where one side was in contact with 3% solution of NaCl and the other with the 0.3 M solution of NaOH, as given in Figure 2.

In the six-hour test, the initial current and that of every 30 min were recorded. During the test, the temperature of the sample or solutions should be controlled. Since electrical current is the amount of passing charge over time, the overall passing charge can be calculated as a definite integral of current in time (software and datalogger measurement) or according to Equation (3) (numerical calculation by the trapezoid method). According to the passing charge result, permeability and thus the quality of cement paste can be evaluated.
(3)$$Q=\int_{0}^{t}I.dt\;=\;900.\left[I_{0}+2.\left(\overset{n}{\underset{i=30}{\sum }}I_{i}\right)+I_{360}\right];\;n\in \left\{60,\;90,\;\ldots ,\;330\right\}\;min$$
where,*Q*—overall passing charge (C);*I*—measured electrical current (A);*t*—time (min).

### 2.7. Thermal Analysis

Thermal analysis was primarily used to determine the content of the main cement composite’s components (ettringite, C-S-H phase, portlandite, and remnant hydration phases). Cement paste samples were dried (at 105 °C), crushed, ground, and analyzed in powder form using a STA Jupiter 4 thermal analyzer (Netsch, Germany) under nitrogen atmosphere at a heating rate of 30 K/min in a temperature range of 25–900 °C in DSC/TG mode. The same mass of samples of 30 ± 1 mg per sample was heated in corundum pots. The contents of the ettringite, C-S-H phase, portlandite, and other hydration phases were detected from the amount of decomposed matter, derived from the particular TG curves, in the temperature range of 105–160 °C, 160–423 °C, 423–500 °C, and 500–900 °C, respectively. Consequently, free portlandite amount, chemical-based water content, and hydration degree were calculated for the individual cement paste samples.

## 3. Results and Discussion

### 3.1. Workability

The results of the workability test are reported in Table 3. Workability is the very first and one of the most important properties of cement composite, since it determines the labor required to place it in a framework [70,71,72]. In the case of unsatisfactory workability, there is the tendency to adjust it with water, decreasing the durability of cement composites [73]. The results of Haegermann’s test, which is based on measuring flowability under dynamic conditions, provides good information, not only on workability/consistency, but also on overall rheology [74].

Difference in consistency between the mixtures with w/b = 0.4 (REF_0.4 sample) and w/b = 0.5 (REF_0.5 sample) 5 min after mixing is illustrated in Figure 3.

The need to adjust consistency (increase in plasticizer dosage) grows with w/b decrease. It is due to better dispersion of binder grains in a mixture with higher water content [13]. This fact is crucial in mix designing, since there is a specific dosage for each plasticizer to achieve specific consistency with a well-designed aggregate composition and at a given w/b ratio [75]. When maximum dosage exceeds the given w/b ratio, no improvement in cement composite consistency is observed; moreover, bleeding occurs [76]. Another observation connected with plasticizer–consistency relation was that using high dosages of plasticizer can lead to higher stickiness (viscosity), which should be evaluated by rheometry.

Replacement of 7.5% of cement by any of the additives, silica-fume and zeolite, (NZ7.5_0.3, NZ7.5_0.4, NZ7.5_0.5, SF7.5_0.3, SF7.5_0.4, SF7.5_0.5) did not have a significant effect on consistency. Replacement of 15.0% of cement (NZ15.0_0.3, NZ15.0_0.4, NZ15.0_0.5, SF15.0_0.3, SF15.0_0.4, SF15.0_0.5) led to deterioration in flowability. The same effect was observed in [77,78]. Thus, the SCMs used could be used as stabilizers in fresh cement composites against segregation and bleeding [79].

No significant decrease of flowability in time was observed in any of the mixtures. On the other hand, adding an admixture at higher dosages (mainly SF7.5_0.3 and SF15.0_0.3 and temporarily NZ7.5_0.3–30 min) led to the “additional plasticization” effect. A mixture of NZ7.5_0.3 for 30 min and SF7.5_0.3 had a self-consolidating character.

In contrast, the highest dosage with a NZ15.0_0.3 mixture was necessary due to these mutual factors:


(a)high amount of replacement by zeolite [56,80];(b)the structure of zeolite (clinoptilolite) crystalline grid-cavities, interconnected by channels, enables water to be stored, leading to a decrease in consistency [81];(c)a less amount of water (w/b = 0.3) [82]. The same is also seen when the results for zeolite and silica fume are compared. SF grains were not fully de-agglomerated and acted as the ball bearings and filler. This also impacted other properties, as will be discussed further. Therefore, the w/b ratio for SF had to be replaced by the w/c ratio, which is higher.


### 3.2. Density

The values of the determined densities of the studied fresh mixtures (D-mix) and prepared samples after 28 and 180 days (D-28-ssd, D-180-ssd, D-28-d and D-180-d) at a particular time and under moisture conditions are given in Figure 4 and Figure 5.

The values of density in fresh state (D-mix) varied from 2078 ± 13 kg m^−3^ (REF_0.3) to 1772 ± 14 kg m^−3^ (SF15.0_0.5). The density of a mixture depends mainly on the densities of its constituents. Therefore, the densities of the prepared cement pastes decrease with w/b ratio as well as with the amount of additive replacement.

The same pattern as for the D-mix is preserved for D-28-ssd, D-180-ssd, D-28-d, and D-180-d. The highest values are achieved for REF_0.3 (2132, 2151, 1879 and 1909 kg m^−3^), the lowest values for NZ15.0_0.5 (1864, 2054, 1733 and 1774 kg m^−3^), and for SF15.0_0.5 (1857, 1854, 1405 and 1434 kg m^−3^). An interesting trend is seen, where generally: (a) density increases in time, even when fresh and hardened state are compared [83], and (b) density increases in ssd as well as in oven-dried state. Despite the lack of literature concerning this issue, the increase of density in time was documented in [84,85,86]. This can be clarified by curing the specimens under water and by the in-building process of water into the silicate matrix within the hydration reaction.

Based on the determined D-mix results, real compositions of each mixture could be calculated. Usually, a volume method—based on first calculating, then measuring, and then correcting—is applied to the mix design precision [87,88]. Using this method, it is important to know the accurate values of the input materials’ specific gravity, to estimate air volume. After fresh density and real air content measurements, it is necessary to correct mix composition with a correcting factor. Our proposal is to apply the simplifier approach—it should consist of choosing the w/c ratio and cement paste/aggregate ratio, then measuring the density of the mixture, and finally calculating as per Equations (4)–(7):(4)$$D_{mix}=m\left(cp\right)+m\left(agg\right)=m\left(c\right)+m\left(w\right)+m\left(agg\right)$$
(5)$$D_{mix}=m\left(c\right)+m\left(c\right)\cdot x+\left(m\left(c\right)+m\left(w\right)\right)\cdot y$$
(6)$$D_{mix}=m\left(c\right)\cdot \left[1+x+\left(1+x\right)\cdot y\right]=m\left(c\right)\cdot \left(1+x\right)\cdot \left(1+y\right)$$
(7)$$m\left(c\right)=\frac{D_{mix}}{\left(1+x\right)\cdot \left(1+y\right)};\;m\left(w\right)=m\left(c\right)\cdot x;\;m\left(agg\right)=\left(m\left(c\right)+m\left(w\right)\right)\cdot y$$
where,*m(cp), m(agg), m(c),* and *m(w)*—mass of cement paste, aggregate, cement, and water, respectively;*x*—water to cement ratio;*y*—cement paste to aggregate ratio.

When this ratio method for CC designing is used, the measured D-mix helps calculate the composition for 1 m^3^. Moreover, composition (primarily w/b ratio) should be re-evaluated after a hardened CC performance. This simplified but effective approach is analogous to concepts of k-value and equivalent performance [89]. The corrected real compositions of the studied mixtures are presented in Table 4.

In the calculation, two situations within mineral additives can be considered. First, when the additive acts as the active, then the w/b ratio should be taken into account (analogous: 0 < k-value ≤ 1). If not, the w/c ratio is taken into account and the additive is considered an inert filler (very fine aggregate; analogous: k-value = 0). This can purely be reviewed by another test, as will be presented further. The calculated cement and water content negligibly differs from the values of the volume methods (about 0.6%). We consider the ratios method to be more undemanding and quicker than the volume one, thus being in compliance with Ernst Mach’s modification of Ockham’s razor in science. The most suitable method in targeted designing of CC mix compositions with required parameters (durability and consequently sustainability) is a combined method. However, particular functions and properties of the filling and binding phase of CC have to be considered.

The values of water content given in Table 4 were used to estimation of non-evaporable water (hereinafter w(non-ev.)) [90,91] in particular composites. W(non-ev.) was calculated as the ratio of the remaining water in samples after drying and binder content, according to (Equation (8)). In the case of silica-fume, w(non-ev.)/c ratios were also calculated, as per the argument stated above. Non-evaporable water content and its development in time are given in Figure 6.
(8)$$w\left(non-ev.\right)=m\left(w\right)-\left(D\left(ssd\right)-D\left(d\right)\right)$$
where,*m(w)*—content of water in fresh CP (kg·m^−3^);*D(ssd)*—density of CP in saturated-surface-dry state (kg·m^−3^);*D(d)*—density of CP in oven-dried state (kg·m^−3^).

It is obvious that with increase of w/c, w(non-ev)/b ratio also increases, which could be explained by more extensive water-cement grain contact in a matrix with higher porosity and permeability and thus rapid ratio of water incorporation into the matrix. Except for silica-fume mixes and mix NZ15.0_0.3 (this we consider to be a random error), a trend of reducing curve steepness can also be seen. We assume that it is caused by excess water, since this increases the rate of hydration. In mixes NZ15.0_0.3, NZ15.0_0.4 and NZ15.0_0.5, this curve may be shifted to the right (or down), due to retaining of water in the zeolite crystalline grid [81] when a higher amount of replacement is used. There is no clear tendency in relative increase of the w(non-ev.)/b ratio relating either to the w/c ratio or to type and amount of replacement; however, we predicted that relative increase should decrease in time and this decrease should be faster with increasing w/c ratio, while with higher w/c, maximum w(non-ev.)/b will be faster reached. This value may correspond to 0.24 (complete hydration of cement [92]). However, this should be subject to more targeted and rigorous experiments. In any case, the determination of chemically bounded water by the described method could be a simple way to control the hydration process in cement composites.

### 3.3. Flexural Strength

The results of flexural strength in different curing times and moisture content of the samples are given in Table 5.

Within some series, it was necessary to reject results that were deviating (maximum one of three per mixture/time/curing condition). High variation factors are due to brittleness of the cement paste composite, where there is a deficit of filler (sand, coarser aggregate), which puts up a resistance against fracture, as in the case of mortar or concrete. Excluding samples REF_0.3 (mainly due to density), the shape of the fracture curve and cross-section area was not linear, but rotund (Figure 7a,b). Moreover, the samples after oven-drying evinces higher brittleness, leading to irregular (zigzag) fracture shape (Figure 7c,d). Although the flexural strength of cement paste is not a crucial property, it could provide important data and be used to fractionalize the beam for compressive strength.

According to the results and the statements declared above, the absolute values of flexural strength are not as important as the following trends, which can be observed. Decrease in f-flex is connected with a decrease in w/b ratio and density [93]. This does not fit samples with SF, since SF acts in this case as a filler, as can be seen in Figure 8. A partial increase in time was observed only for the reference REF_0.3-REF_0.5 samples in ssd state.

The values of samples in oven-dry condition are significantly lower than those in ssd state, which could have been caused by an increase in brittleness after the drying process [94,95]. A decrease in f-flex of cement pastes at 100 °C was also reported in [96]. In addition, the results of individual oven-dried specimens of the samples vary more than those in the ssd state.

### 3.4. Compressive Strength

The results of compressive strength on day 28 and day 180 for each moisture state are given in Figure 9.

Mutual comparisons between particular results are given in Figure 10.

If there are several alongside results at the extreme end, they indicate the same means of population according to unpaired two-sample Student’s t-test at a significance level of 0.05.

The results of fc-28-ssd vary from 40.7 ± 2.2 (SF7.5_0.5 sample) and 40.9 ± 3.7 (SF15.0_0.5 sample) to 103.1 ± 5.5 (the REF_0.3 sample) and 107.5 ± 3.1 MPa (NZ7.5_0.3 sample). The values of fc-28-d fluctuate between 45.2 ± 3.1 (SF7.5_0.5 sample) to 138.9 ± 8.3 MPa (REF_0.3 sample). Compressive strength in saturated state after 180 days (fc-180-ssd) ranges between 58.6 ± 2.6 (SF15.0_0.5 sample), 59.2 ± 4.1 (NZ15.0_0.5 sample), and 61.2 ± 1.9 (SF7.5_0.5 sample) at the minimum and 109.9 ± 5.1 (NZ7.5_0.3 sample) and 115.6 ± 5.4 MPa (REF_0.3 sample) at the maximum. The results of fc-180-d fluctuate in the range from 54.9 ± 2.0 (NZ15.0_0.5 sample) and 55.1 ± 3.2 (SF7.5_0.5 sample) to 142.6 ± 5.3 MPa (REF_0.3 sample).

Compressive strength decreases with w/b increase, which is well-known as Abram’s law [93,97]. This is also valid for samples with SF, when the w/b ratio is substituted by the w/c ratio, as given in Table 4. Re-calculation had to be performed due to visual evaluation of cross-section of samples with SF (Figure 8) and due to the assumption that SF should have a much higher activity [55] than observed. The same observation about compressive strength of CP with densified silica-fume is provided in [98,99]. The reason for this is because no-deagglomeration occurred while mixing, as cement paste does not have grounding capability. Relations within our experiment are given in Figure 11, where the coefficient of determination reaches 0.9743.

Compressive strength was confirmed as a reliable variable in active additive assessment, but is not the only one, while f_c_ cannot differ between active additive and high absorbance fineness. Slight decrease of f_c_ can be observed when a higher amount of replacement is used, which is in line with [100,101,102].

Considering relation f_c_-180-ssd:f_c_-28-ssd, there is an obvious trend, since compressive strength increases with time, complying with theory. Moreover, relative increase is raised with w/b ratio. This phenomenon can be observed on mortars f_c_ result in [103]. This means that the f_c_ of mixtures with lower w/b ratio increases rapidly at the beginning and then more slowly, compared with a higher w/b ratio, where the development is gradual. In practice, it is called “cement burnout” and can constitute a problem in quality control management, since seven-day compressive strength is considered an indicator of strength, which, however, does not need to increase to the required 28-day compressive strength.

Excluding samples NZ7.5_0.3 and NZ7.5_0.4, oven-dried samples achieved significant higher values (increase from 10.5 rel.% up to 48.9 rel.%) of compressive strength on day 28 day, as those in ssd state. This has been proven in our previous studies [21,104].

The property stated above is not convincing for day 180, nor when f_c_-180-d and f_c_-28-d are compared. According to [105,106,107], we strongly presume that compressive strength is intensively affected by the moisture state of the sample at the time of performing the test. Moisture content probably has more effect than curing conditions. This also should be a subject of extent and rigorous research with respect to durability and consequently sustainability, since we could highly underestimate real compressive strength.

The relationship between compressive strength and water absorption, which is given in Figure 12, is also obvious, with a coefficient of determination equal to 0.8749. This is due to the fact that compressive strength as well as water absorption are linked to porosity [108].

### 3.5. Rapid Chloride Penetration Test

The values of the overall charge (Q) that passed through the 180-day samples over a 6-h period are given in Figure 13.

For comprehensiveness, values of initial current, maximum current, and current at the end are also presented. The values of charge varied from 433C (SF15.0_0.3 sample) to 4091C (REF_0.5 sample) without conversion to the d = 95 mm samples. It is obvious again that durability, in this case represented by permeability, decreases with w/b increase. The second trend that can be seen is permeability decrease with amount of cement replacement. As for compressive strength, the NZ used is a convenient replacement from a permeability point of view, but an “improver” for cement composites [109]. The samples with SF should be considered as a separate case, as despite a higher w/c ratio compared to REF_0.3-REF_0.5 samples, they reached lower values of passing charge. This could be because SF acted as a filler and barrier for penetration, while breaking the pore connection. The confirmation for this could be because mortars achieve lower Q in the RCP test as well as concretes in comparison with mortars [110]. In addition, partly pozzolanic reaction of SF grains surface will lead to permeability improvement [111].

This measurement technique is considered to be determinative, and also should be declared as the rapid ions passing test (RIPT), while other ions like OH^-^ from the sample intensify passing charge [69]. The first consideration is based on the temperature effect that accelerates passing ratio, adhering to Fick’s first law and Nernst-Planck equation. The more permeable the sample is, the more charge passes though, a higher current is achieved, and then higher Joule (Ohm) heat is developed and temperature rises [68]. This heat is divided between the electrolytes solution (lower part) and the sample (higher part), as per the thermal capacity of the material. Consequently, poor quality (in sense of high permeability) CC are classified as worse than they real are [112]. Temperatures within permeable samples (w/b > 0.5, 28 days) can exceed 100 °C right after test initialization; therefore, cooling is necessary. However, according to our observations, when the sample sealing is not deteriorated, maximum current is reached and the current development curve is broken (current starts to decrease), and temperature starts to decrease. A connection between the initial current I(0) overall charged passed (Q) can also be observed. Generally, low initial current foreshadows low total charged [113]. On that account, we propose, relating to this test (mainly when cement pastes are subjects of research), values of initial, maximum, and final current, respectively, and temperature measurement. This can be replaced by combining all these variances into one plot.

Three measurement progressions from our experiment are given in Figure 14a (without temperature progress, which in our case served only for controlling the electrolytes due to safety). A schematic detail of current fluctuation within the RCPT is illustrated in Figure 14b.

As can be seen in Figure 14b, periodic fluctuation of moving current was observed, when it slowly decreased and rapidly increased. The rate of fluctuation increased with more permeable samples. From this point of view, the sample acted like a capacitor. Typical current progression of more permeable samples has been observed to consist of three phases:


(1)rapid increase of current and temperature,(2)stabilization of maximum increase of temperature,(3)slow current and temperature decrease. These findings can be confirmed by values of initial, maximum, and final current values, while these variances are more disarranged in the case of high permeable samples than those with low permeability.


### 3.6. Water Absorption

The results of water absorption after 28 days (WA-28) and 180 days (WA-180), as well as relative decrease of water absorption in time are given in Figure 15.

The values of WA-28 of the samples ranged from 15.1 ± 0.3 wt.% (REF_0.3 sample) to 35.6 ± 0.2 wt.% (NZ15.0_0.5 sample) and for WA-180 from 14.5 ± 0.2 wt.% (REF_0.3 sample) to 34.0 ± 0.1 wt.% (NZ15.0_0.5 sample). WA decrease over time can also be observed. This could be explained by microstructure densification due to the hydration process [114]. An obvious dependence between w/b ratio and WA was confirmed, while WA increased with the amount of free water that can be evaporated from the composite [115,116,117]. When we take into consideration the fact from our assumption that the real w/b ratio for all the mixtures SF7.5_0.3-SF15.0_0.5 are in fact equal to the w/c ratio (Table 4), the question arises: why are the WAs of these mixtures not higher? The reason for this is most likely because we should see the WA of the final composite as a result of the WAs of its particular components. In mixtures SF7.5_0.3, SF7.5_0.4, and SF7.5_0.5, the amount of SF represents 5.8, 5.3, and 5.0 wt.%, respectively and in mixtures SF15.0_0.3, SF15.0_0.4 and SF15.0_0.5 it is 11.5, 10.7, and 10.0 wt.%, respectively. The WA of densified SF that is not deagglomerated is much lower than that of cement.

In Figure 8, it can be seen that only a little part of the SF grain is penetrated by water. Higher w/c ratios of a lower amount of cement paste within the SF samples are compensated by lower WA of SF, and the resulting WAs are similar to that of other mixtures (REF and NZ ones)) within a particular w/b ratio. This procedure could be used as a technique to assess activity of the additive. Our proposal to design a mix composition for the final WA is based on Equation (9) and Figure 16. Basically, the equation represents the weighted average of water absorptions of cement paste and aggregate as separate phases.
(9)$$WA\left(cc\right)=\frac{m\left(f\right)}{D\left(cc\right)}\cdot WA\left(f\right)+\frac{m\left(cp\right)}{D\left(cc\right)}\cdot WA\left(cp\right)$$
where,*WA(cc)*—overall cement composite water absorption (wt.%);*m(f)*—filler (aggregate) content in fresh cement composite (kg·m^−3^);*D(cc)*—density of cement composite in fresh state (kg·m^−3^);*WA(f)*—water absorption of filler (wt.%);*m(cp)*—cement paste content in fresh cement composite (kg·m^−3^);*WA(cp)*—water absorption of cement paste (wt.%).

From our experience, when we retroactively submitted various CC compositions (not yet published), the precision of the proposed equation reached >95%. The significance of the proposed calculation was based on predetermination of the final WA, while CC composition is designed, mainly for environment exposure class XA. In addition, it can serve as a tool for inspection of (a) retrospective determination of mix composition (mainly cement content and w/c ratio) or (b) determination of WA result plausibility. In Slovak Republic, the maximal level of WA is 6.0 wt.% on the 28th day for exposure classes XA1-XA3, as per STN EN 206+A1 [118]. According to the results presented in this study and assuming that (a) the mass of cement paste in CC is 25% (equals 28 vol.%), (b) WA of the aggregate is 1.0 wt.%, then the limit would be met with w/c < 0.40. Therefore, we are strongly convinced that the limits for maximum w/c (XA1-0.55; XA2-0.50; and XA3-0.45, respectively) or for WA are inappropriate according to the stated standard. The problematic factor in WA determination of the aggregate according to EN 1097-6:2013-07 [119], is due to particle <0.063 mm washing out. Thus, the WA value of the aggregate is lower than that with the most absorbent filler. Consequently, in CC, these particles were able to absorb mix water, decrease w/c ratio, and thus result in the overall composite’s WA. Therefore, we should determine the WA of aggregate with particles <0.063 mm or WA of cement paste, including these particles.

In general (excluding NZ15.0_0.3), the increase in relative decrease of WA with time relates to a higher w/b ratio. When the shape of the curves is compared with this in Figure 6, an apparent similarity is seen, since WA and D are based on a similar measuring technique and with properties they are connected to.

### 3.7. Thermal Analysis

Mass loss corresponding to decomposition of the main hydration components, derived from the TG curves, are presented in Table 6. Chemically bound water and hydration degrees were calculated for the 28-day and 180-day cured samples according to [120]. TG/DTG curves of the NZ15.0_0.3 sample are given in Figure 17.

The content of C-S-H and portlandite phases increased with time for all the studied samples. This confirms the ongoing hydration processes in the cement samples. The absolute 180-day content of C-S-H increased with the w/b ratio more significantly than that of portlandite. On the contrary, the amount of hydration phases decomposing above 500 °C decreased for the samples with additives, when compared the 28-day and 180-day values.

On comparing the pastes with SCMs of various replacement ratios, it can be concluded that the content of portlandite decreased with increase in the additives’ ratio. For both NZ- and SF-blended composites, the content of portlandite was lower in samples with 15 wt.% replacement of cement than in samples with 7.5 wt.% replacement. This is likely caused by more effective pozzolanic activity at a higher amount of SCM as a pozzolanic additive. However, this was not confirmed by the C-S-H phase increase in samples with higher cement substitution by SCMs than expected.

Chemically bound water increased with time and varied from 12.7 to 16.4% and from 13.4 to 17.9% for the 28-day and 180-day samples, respectively. The lowest relative increase in chemically bounded water was observed for the NZ15.0_0.3 and NZ15.0_0.4 samples (2.9%) and the highest one for the REF_0.5 sample (16.7%). The chemically bound water values determined by TA (w_ch.b._) are in a good correlation with the calculated w(non-ev.) values of the reference and composites with mineral additives, determined according to density as presented above. The fact that samples for TA were pretreated by drying at 105 °C has to be taken into consideration, since many cement hydrates like C-S-H, ettringite, and monosulfate, lose part of their chemically combined water below 105 °C [121]. Having said that, distinguishing between chemically (or non-evaporable) bound water, physically adsorbed, and free water (together as evaporable water) is a complex issue [89]. On the other hand, computation as per the densities presented in this paper could be used as a simple control method.

Based on Batthy’s method, the degree of hydration α is directly proportional to the content of chemically bound water. Figure 18 presents the relative changes in hydration degrees over time per individual samples.

Obviously, the hydration degree increased with time. In Figure 18, two patterns can be seen. First within the NZ7.5 and SF7.5 samples (concave shape) and second within the NZ15.0 and SF15.0 samples (convex shape). Regarding the sensitivity of hydration degree determined from TA and techniques for its investigation, it will be necessary to perform more extent and rigorous research.

### 3.8. Evaluation of Mineral Additives and Their Effect on the CC with Regards to Durability and Environmental Suitability

Every new mineral additive that is intended for use in a cement composite should undergo comprehensive analysis. It could then be categorized as per its activity. The proposed evaluation model is based on a comparative analysis of tests covering mechanical, physical, microstructural, and chemical properties. For effective application of the proposed model, reference cement pastes should be prepared. For optimal w/c ratio for the investigated cement paste with and without mineral additive, we consider 0.4 (w/c ref). In addition, reference cement pastes without the subjected mineral additive with higher (w/c–up, e.g., 0.5) and lower (w/c–down, e.g., 0.3) water to cement ratio should be prepared. Then, each result should be compared (a) firstly to w/c ref, (b) then to w/c down or up ratios. The outcome of the performance [35] analysis of the particular results should lead to categorization into one of the following four proposed mineral additive types (summarized in Table 7).

A type—inert material with low water absorption (for example, coarse grain additives having water absorption values similar to sand, approximately up to 2.5%). This mineral additive would play a role of fine filler (but coarser than the proposed B type material) in CC. This does not mean that after grinding, it becomes a cement supplementary material (as SF in this study). When using this type, it is assumed that the real w/c ratio of CC will increase due to unabsorbed water in the subjected material or because a small surface is exposed to the hydration reaction. This likely worsens durability and thus sustainability of CC.

B type—inert material with high water absorption (for example, a dust or filler with no pozzolanic or latent hydraulic properties like crystalline fine limestone or quartz with water absorption values up to 20%). It is assumed that if applying this kind of mineral additive, the real w/c ratio of CC will decrease due to physically bound water in the material. This additive could act as an improving (but not active) additive and its incorporation could result in indirect improvement in some CC durability properties. This is due to the filler effect and physical stimulation of cement hydration [122]. We consider the use of the term SCM in [123] for this type of mineral additive as controversial.

C type—material similar to cement. SCM has pozzolanic or latent hydraulic properties that are objectivized by mineralogical analysis [121,124]. The overall performance of the CC with this additive type should be at least the same or at a slightly better level as the reference CC without SCMs.

D type—the subjected material as an improving active additive. This has pozzolanic or latent hydraulic properties that are objectivized by mineralogical analysis. The overall performance of the CC with this additive type should achieve significantly better value of at least one tested property than the reference paste. We use this distinction (C and D types) only due to semantic precision. Furthermore, the amount of replacement plays a key role, since some mineral additives act as SCM only at specific replacement ratios [125].

We highlight that the results have to be compared to the reference cement paste in accordance with statistically unpaired two-samples test, due to insufficient individual results that are usually obtained. It is also necessary to perform a long-term investigation in connection with the development of durability properties over time [126].

#### 3.8.1. Prediction of Consistency

Rheology as a set of properties in fresh state is the most crucial aspect of CC in practice [127]. When an adjustment of workability is performed on the construction site, it could lead to durability deterioration (when additional water is used) or risk of segregation (when a plasticizer is used) [128]. Thus, robust composition of cement paste with suitable consistency is necessary to know when the CC is designed [129]. With the assessment of a mineral additive, two options may occur when a consistency test is performed. The first one—a higher consistency than for the reference paste will be typical for A type, due to a higher w/c ratio, since the mix water is only partially absorbed into the material. The second option—a lower or the same consistency as for the reference paste will be observed for the B, C, and D types, respectively, since consistency usually becomes stiff with active additives, as well as fineness (Figure 19) [49,50,55,130,131]. Water demand and thus change in workability is mainly based on the mineral additive surface area [132].

#### 3.8.2. Prediction of Compressive Strength of CC

The curves representing the compressive strength trends for particular mineral additive types, based on our experimental research, are presented in Figure 20. When type C represents a trend for the reference paste (C is assumed to follow w/c ref *ex definition*), CC with A type additive should follow the compressive strength curve “w/c up” and the values of the compressive strength will be lower than those for the reference paste. Contrariwise, CC with B type additive is predicted to follow the “w/c down” curve (Figure 20). Cement pastes with an improving active additive (D type) are expected to perform according to the green curve in Figure 20 [103,133]. In the first phase, the compressive strength will likely be under the “w/c ref” curve, which means that the values of compressive strength reach a lower level than for the reference paste without the improving additives; however, in the long-term, the compressive strength will grow and reach higher values compared to the reference composite. However, replacement ratios have to be taken into consideration, since decrease occurs when a high replacement of cement is used [103,134,135].

#### 3.8.3. Prediction of Water Absorption of CC

The water absorption of cement paste is as good as the compressive strength connected to the w/c ratio [117]. Furthermore, when A and B type additives are incorporated into CC, the overall WA is related to the WA of the particular components and their quantity. Thus, CC with an A type additive is assumed to have a little higher WA, and CC with a B type additive, the same or a little lower. In cases of CC with C and D type additives, WA reaches a value as the reference composite (Figure 21); however, its decline over time should be faster [114,136]. The amount of replacement should also be taken into account. Water absorption as a property cannot qualify the material; on the other hand, it is possible to first approximately estimate the w/c ratio of the CC sample and then the amount of cement and water, when filler (aggregate) composition is known.

#### 3.8.4. Prediction of Permeability of CC

Permeability relates to both quality and quantity of the cement paste. The CC with A and B type additives would have a lower permeability due to the number of “flow obstacles” (in A case) [110] and due to a lower real w/c ratio than in the reference paste (B case) [137]. To classify as a C type additive, it is necessary to reach a value at least the same as the reference composite; however, from experience, the value is much lower than that of the reference one [135], even lower when compared to A and B types. The permeability value of CC with a D type additive should be better ex definition; moreover, the same as for the C type is valid in this case. The estimated values are illustrated in Figure 22.

#### 3.8.5. Prediction of Hydration Phases Composition in CC

Quantitative hydration phases analysis (by X-ray diffraction or thermogravimetric analysis) and their time-scale development would be able to distinguish between the filling effect of inert additives (A and B types) and active additives (C and D types) [138,139]. The main phases, C-S-H and portlandite, should be closely investigated with regards to their amount and mutual transformation with time (a more distinguishing aspect) [140]. Their content relates to the quality of cement paste and its components. Cement paste with an A type additive having a higher real w/c as in the reference paste would have a slightly higher content of C-S-H and lower content of portlandite; the development of phases with time would be a little more upward. This can be explained by more space for hydration product formation [141]. In CC with a B type additive, both phases would reach lower content and development would be flatter than in the reference paste. In the case of CC with a C type additive, the content as well as time development should be very similar to the reference cement paste, but as well as within the CC with D type additive, it is assumed that the content of the C-S-H phase should be significantly higher and portlandite lower, while the development ratio should be quicker (Figure 23) [142].

## 4. Conclusions

The study shows the influence of various durability properties by mineral additives. Natural zeolite and silica-fume at doses of 7.5 wt.% and 15.0 wt.% were used as a replacement for cement. The following conclusions can be drawn:


Monitoring of CC density (in ssd and oven-dried state) over time could provide decent information not only about water absorption but also about the hydration process. The chemically bound water, calculated from densities and thermal analysis, evinces good correlation.Flexural strength should not be regarded as a cement paste durability parameter due to its high brittleness, which leads to sudden collapse during loading and thus to low precision of measurements.Despite the fact that compressive strength is the most frequently measured property of CC, for complex evaluation of CC durability, it is necessary to perform additional non-mechanical tests, such as permeability and chemical analysis.The evaluation of results should undergo statistic comparative analysis and Student’s t-test seems to be suitable enough.The moisture content of the specimen significantly affects compressive strength, which could lead to overestimation or underestimation of durability, and thus sustainability of CC.The results of the rapid chloride penetration test could be considered an efficient qualificator of cement paste durability evaluation when mineral additives are assessed.Monitoring the parameters’ progress over time is inevitable in the evaluation of mineral additives’ activity.


This work contributes to a wide research on cement composites’ durability and sustainability. We believe we have fulfilled the goal of this experimental study—to propose the concept of mineral additives’ evaluation in comprehensive analysis of cement paste properties. The concept presented at the end of the discussion has reasonable application capacity in current research and development of cement composite mineral additives mainly due to its holistic approach. Within the presented concept, it is necessary to consider the investigated material as unknown, with regards to its activity or improvement properties. Classification of the investigated additive type can be performed until complex evaluation of the effects on specific CC properties is done.

## Figures and Tables

**Figure 1 materials-14-01448-f001:**
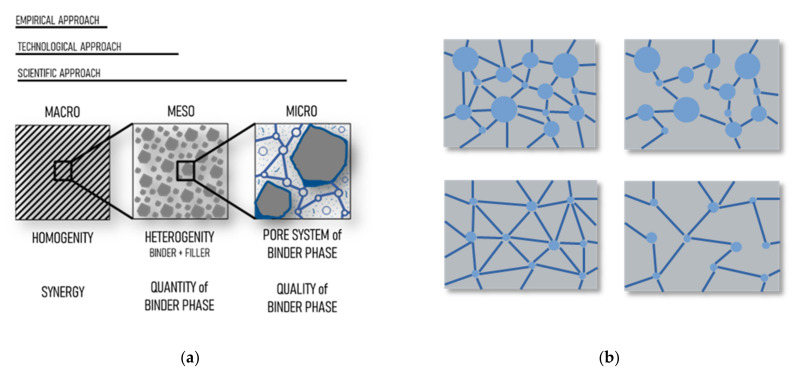
Concept of cement composite designing: (**a**) Multi-scale approach; (**b**) Variations of an open pore system in a cement paste.

**Figure 2 materials-14-01448-f002:**
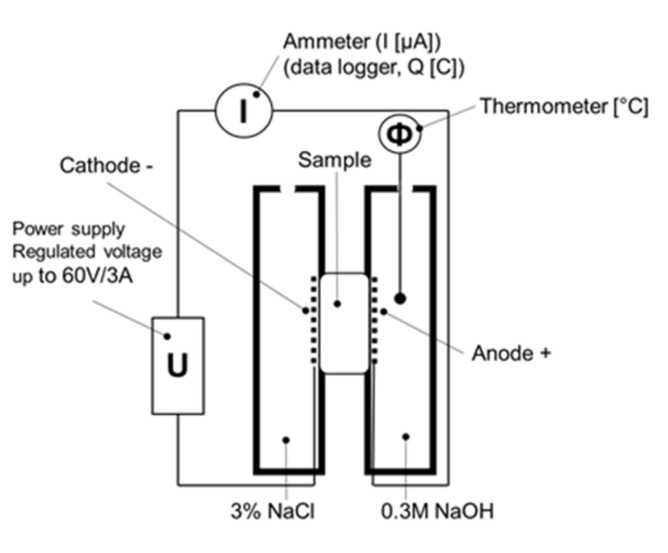
RCPT set-up.

**Figure 3 materials-14-01448-f003:**
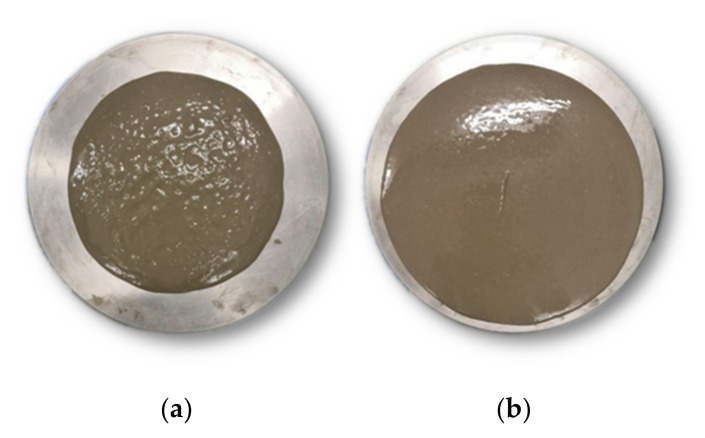
Consistency after mixing (5 min): (**a**) REF_0.4; (**b**) REF_0.5.

**Figure 4 materials-14-01448-f004:**
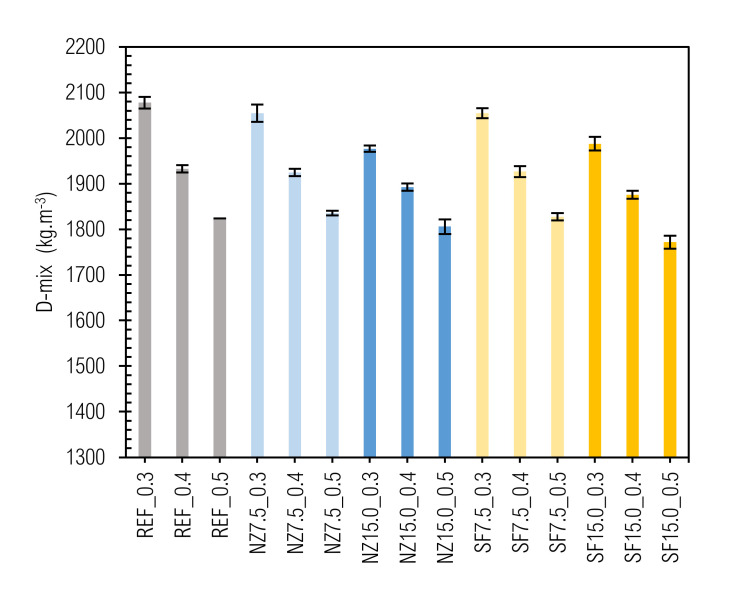
Fresh pastes’ density.

**Figure 5 materials-14-01448-f005:**
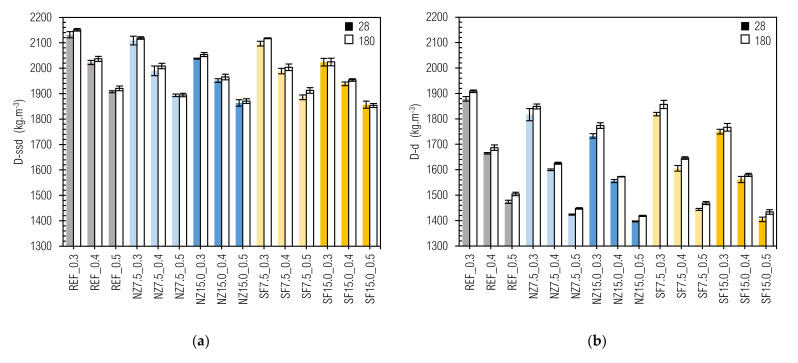
Densities of hardened cement pastes (CP): (**a**) ssd state; (**b**) oven-dried state.

**Figure 6 materials-14-01448-f006:**
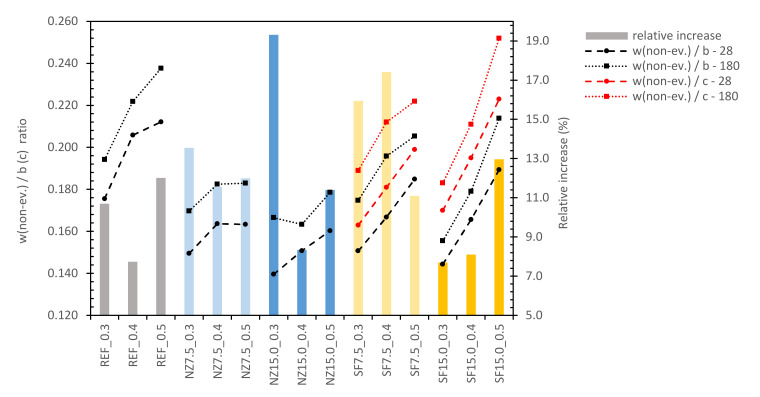
Development of chemically bounded water.

**Figure 7 materials-14-01448-f007:**
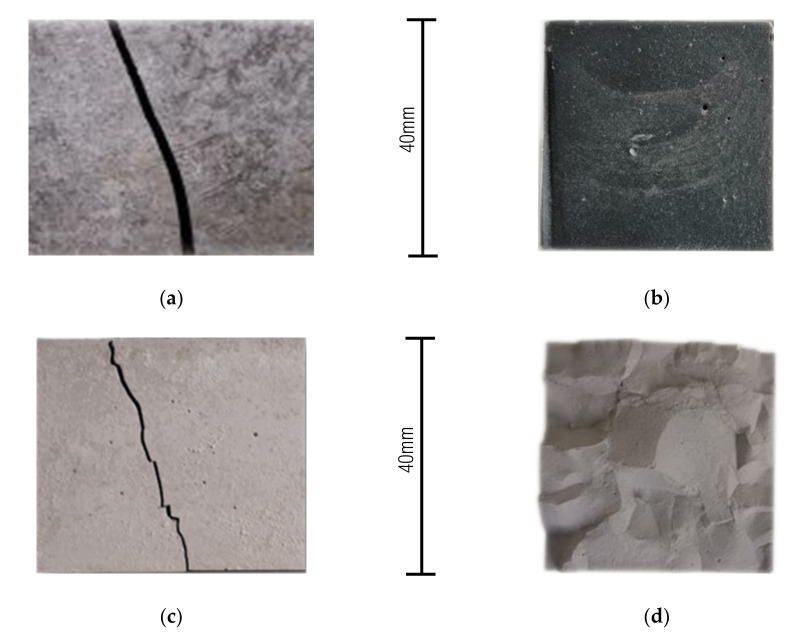
Fracture shape of sample in (**a**) ssd state—curve; (**b**) ssd state—cross-section area; (**c**) oven-dried state—curve; (**d**) oven-dried state—cross-section area.

**Figure 8 materials-14-01448-f008:**

Cross-section and details of samples with SF (silica fume). The needle (black line) indicates an SF grain.

**Figure 9 materials-14-01448-f009:**
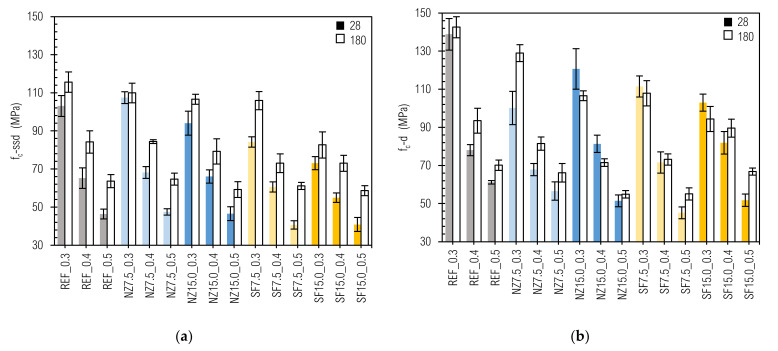
Compressive strengths: (**a**) ssd state; (**b**) oven-dried state.

**Figure 10 materials-14-01448-f010:**
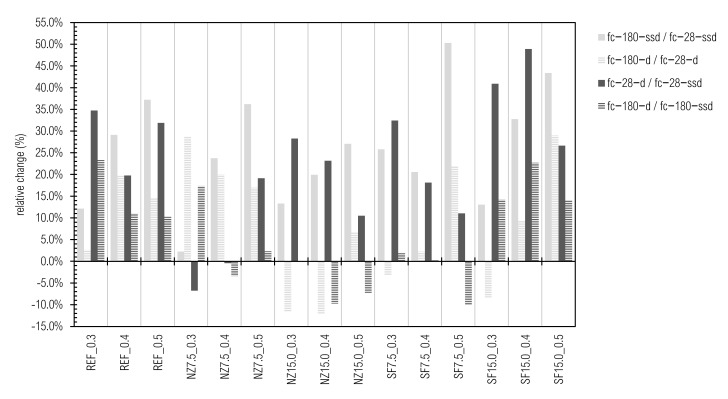
Development of compressive strength and mutual relationships.

**Figure 11 materials-14-01448-f011:**
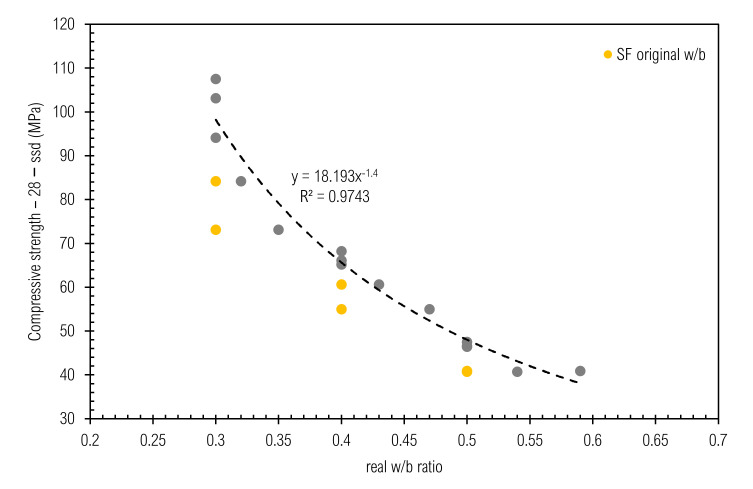
Relation between f_c_−28−ssd and real w/b ratio.

**Figure 12 materials-14-01448-f012:**
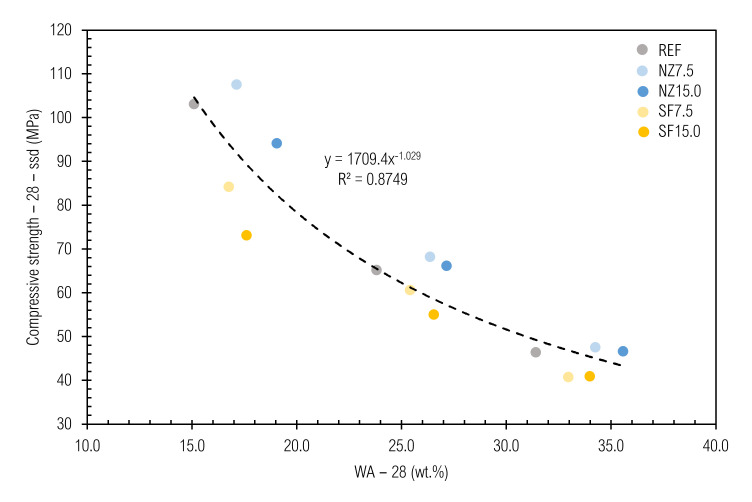
Relation between f_c_−28−ssd and real WA−28.

**Figure 13 materials-14-01448-f013:**
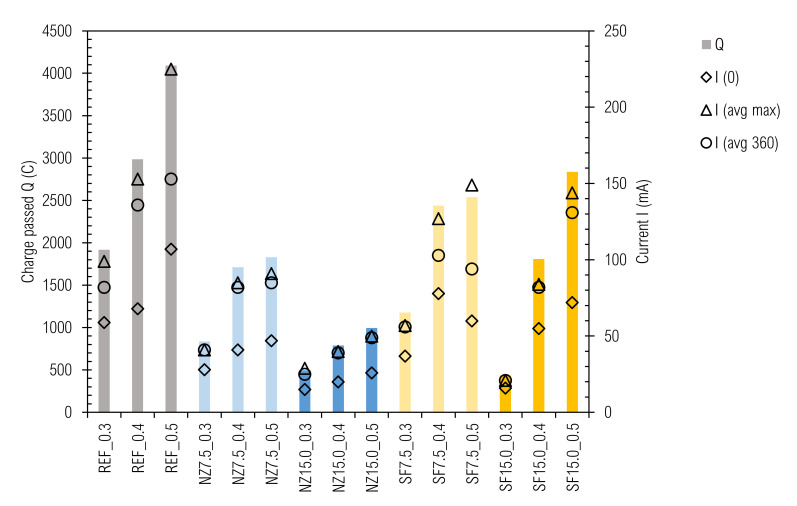
Chlorides penetration test results.

**Figure 14 materials-14-01448-f014:**
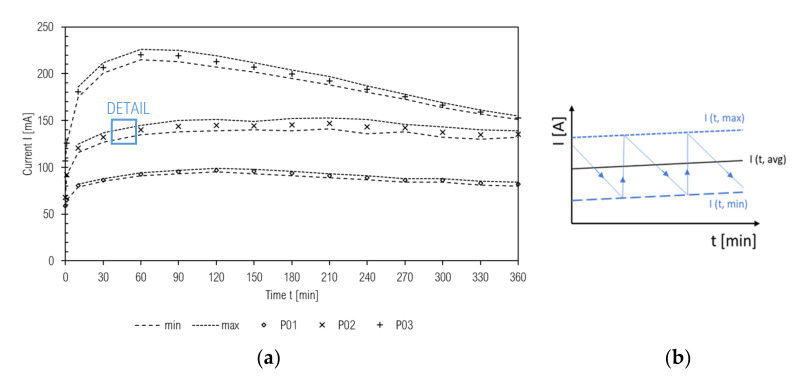
(**a**) Current development in the RCP test; (**b**) schematic detail of current fluctuation.

**Figure 15 materials-14-01448-f015:**
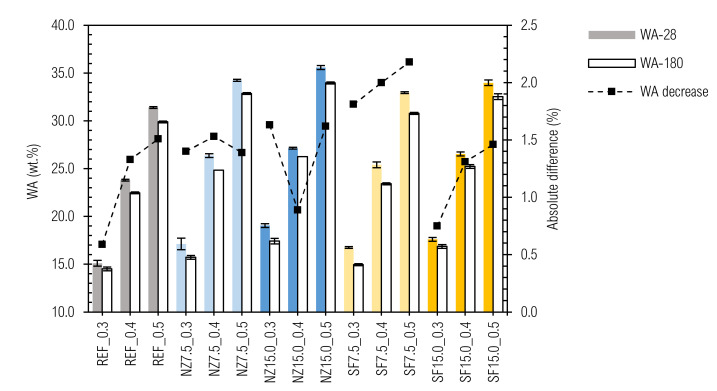
Results of water absorption.

**Figure 16 materials-14-01448-f016:**
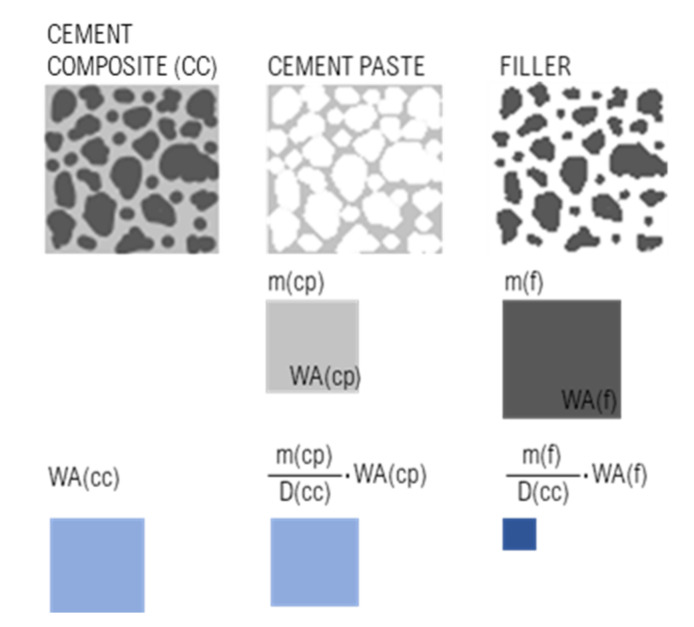
Graphical interpretation of Equation (9).

**Figure 17 materials-14-01448-f017:**
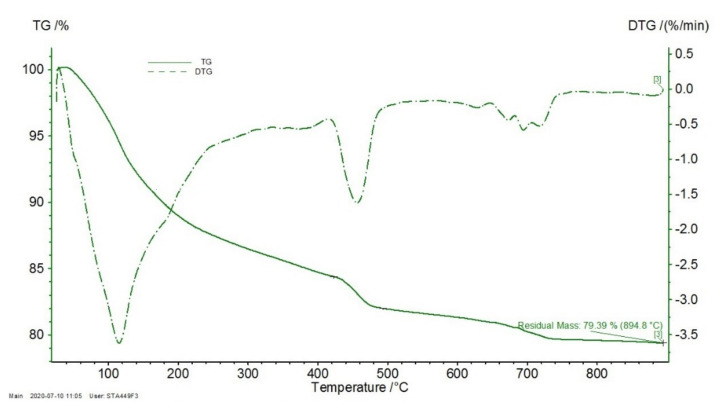
Thermogravimetric (TG)/DTG curves of the NZ15.0_0.3 sample.

**Figure 18 materials-14-01448-f018:**
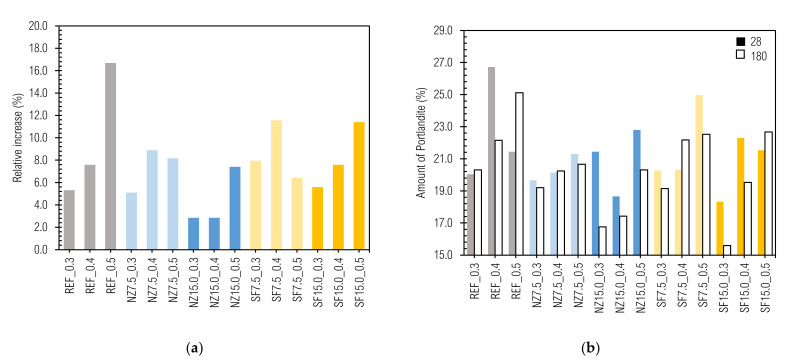
(**a**) Development of hydration degree; (**b**) amount of portlandite on day 28 and day 180.

**Figure 19 materials-14-01448-f019:**
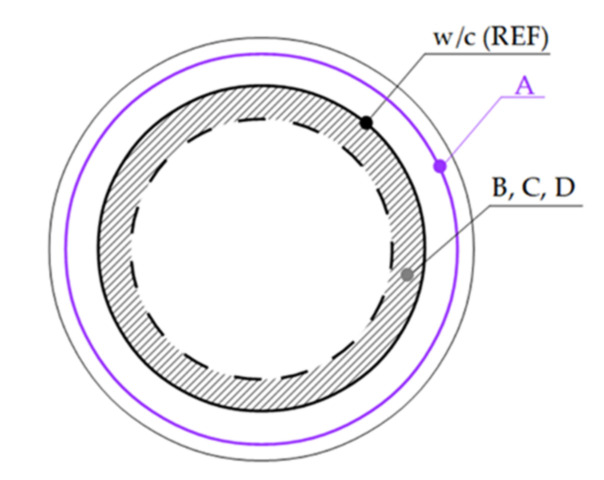
Predicted consistencies (flow diameters on Haegerman’s table) of fresh mixtures depending on the type of mineral additive.

**Figure 20 materials-14-01448-f020:**
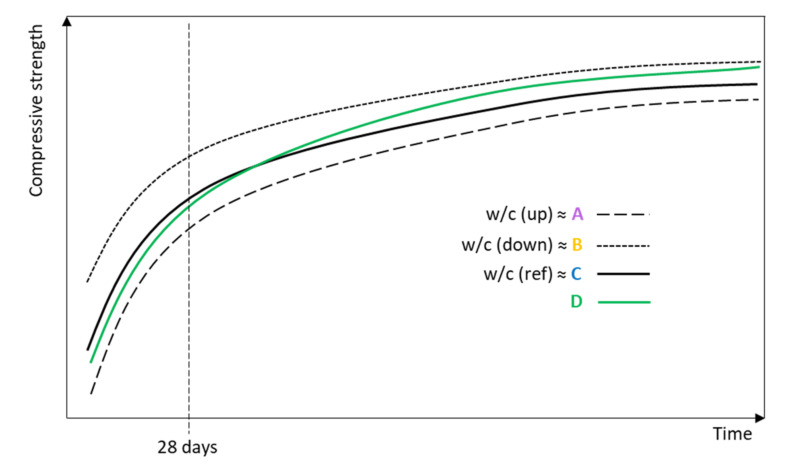
Predicted compressive strengths of CC depending on the type of mineral additive.

**Figure 21 materials-14-01448-f021:**
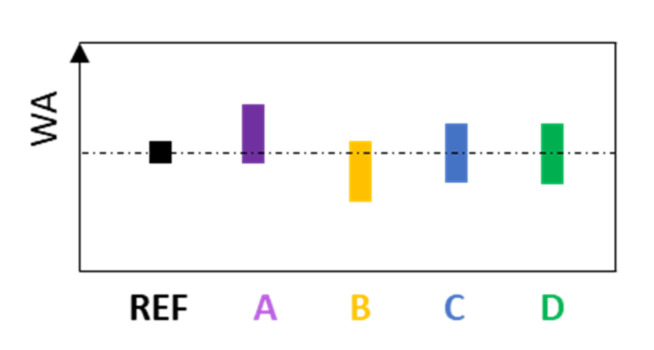
Predicted water absorption of CC depending on the type of mineral additive.

**Figure 22 materials-14-01448-f022:**
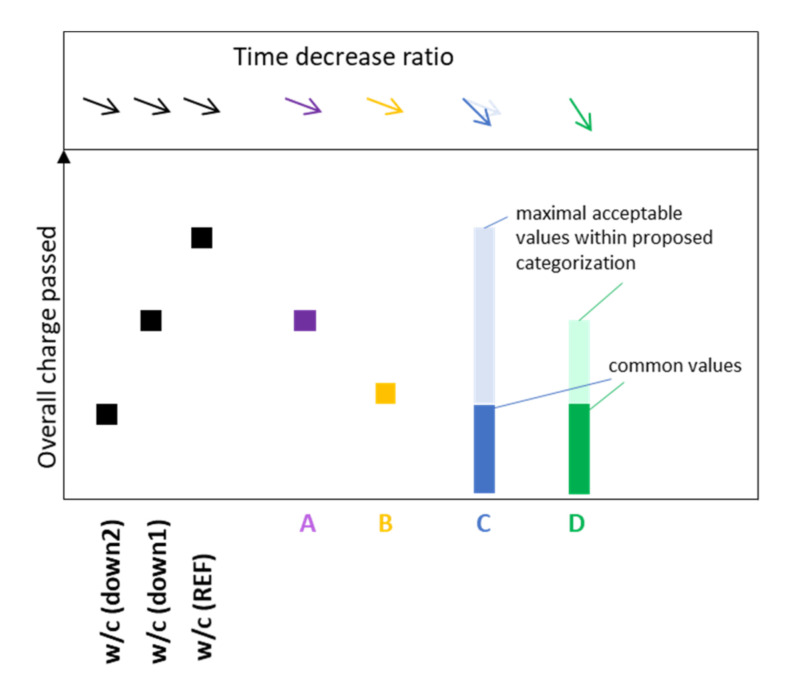
Predicted penetration of ions expressed by charge passing through the CC depending on the type of mineral additive.

**Figure 23 materials-14-01448-f023:**
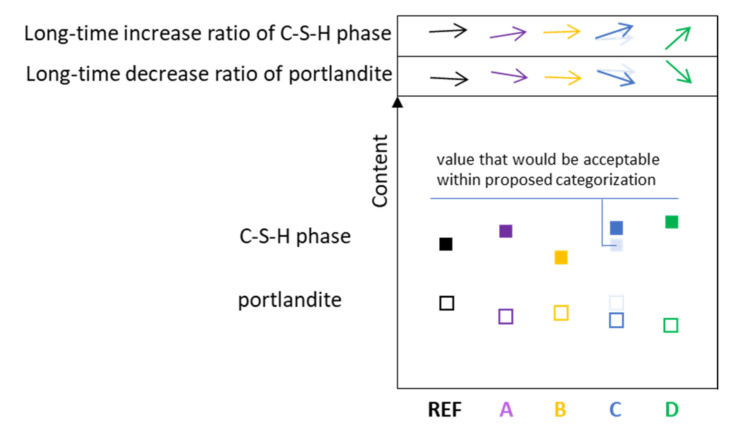
Prediction of the content of hydration phases and their development over time in CC depending on the type of mineral additive.

**Table 1 materials-14-01448-t001:** Chemical composition and specific gravity of the binding materials used.

	Main Elements Expressed as Oxide (wt.%)								Specific Gravity (kg·m^−3^)
	SiO_2_	Al_2_O_3_	CaO	K_2_O	Fe_2_O_3_	MgO	SO_3_	Other	
**Cement**	18.6	3.7	62.1	0.9	2.8	2.1	4.4	5.4	3050
**Zeolite**	58.7	9.0	2.8	2.6	1.4	0.7	0.1	24.7	2440
**Silica-fume**	92.5	0.4	0.3	1.0	1.2	1.0	0.1	3.5	2200

**Table 2 materials-14-01448-t002:** Labeling and compositions of the prepared mixtures.

Component	Replacement (wt.% of Cement)	w/b Ratio		
		0.3	0.4	0.5
Reference	0.0	REF_0.3	REF_0.4	REF_0.5
Natural zeolite	7.5	NZ7.5_0.3	NZ7.5_0.4	NZ7.5_0.4
Natural zeolite	15.0	NZ15.0_0.3	NZ15.0_0.4	NZ15.0_0.4
Silica-fume	7.5	SF7.5_0.3	SF7.5_0.4	SF7.5_0.4
Silica-fume	15.0	SF15.0_0.3	SF15.0_0.4	SF15.0_0.4
Total binder (g)		5600	4800	4200
Water (g)		1680	1920	2100

**Table 3 materials-14-01448-t003:** Results of consistency in Haegermann’s table.

Label	HWR Admixture	5 min	30 min	60 min
	(wt.% of Cement)		(mm)	
REF_0.3	0.34	228	240	247
REF_0.4	0.06	220	240	231
REF_0.5	0.00	262	251	260
NZ7.5_0.3	0.34	270	232 (17) *	252
NZ7.5_0.4	0.06	230	225	218
NZ7.5_0.5	0.00	273	260	258
NZ15.0_0.3	0.70	228	210	192
NZ15.0_0.4	0.06	198	195	192
NZ15.0_0.5	0.00	235	231	237
SF7.5_0.3	0.52	277	243 (14) *	243 (14) *
SF7.5_0.4	0.06	242	213	234
SF7.5_0.5	0.00	281	265	279
SF15.0_0.3	0.52	182	225	223
SF15.0_0.4	0.06	220	218	211
SF15.0_0.5	0.00	255	256	251

* Result without impacts (amount of strikes till maximum limit—300 mm—was reached).

**Table 4 materials-14-01448-t004:** Recalculated mix compositions.

Label	D-mix avg	Water (w)	Cement (c)	SCM Additive (a)	w/c	w/(c + a) = w/b
	(kg m^−3^)	(kg m^−3^)	(kg m^−3^]	(kg m^−3^)	(-)	(-)
REF_0.3	2078	480	1598	0	0.30	0.30
REF_0.4	1933	552	1381	0	0.40	0.40
REF_0.5	1824	608	1216	0	0.50	0.50
NZ7.5_0.3	2055	474	1462	119	0.32	0.30
NZ7.5_0.4	1925	550	1272	103	0.43	0.40
NZ7.5_0.5	1836	612	1132	92	0.54	0.50
NZ15.0_0.3	1977	456	1293	228	0.35	0.30
NZ15.0_0.4	1893	541	1149	203	0.47	0.40
NZ15.0_0.5	1806	602	1023	181	0.59	0.50
SF7.5_0.3	2055	474	1462	119	0.32	0.30
SF7.5_0.4	1927	551	1273	103	0.43	0.40
SF7.5_0.5	1828	609	1128	91	0.54	0.50
SF15.0_0.3	1988	459	1300	229	0.35	0.30
SF15.0_0.4	1876	536	1139	201	0.47	0.40
SF15.0_0.5	1772	591	1004	177	0.59	0.50

**Table 5 materials-14-01448-t005:** Flexural strength results.

	f_flex_-28-ssd	Variation	f_flex_-28-d	Variation	f_flex_-180-ssd	Variation	f_flex_-180-d	Variation
	(MPa)	(rel. %)	(MPa)	(rel. %)	(MPa)	(rel. %)	(MPa)	(rel. %)
REF_0.3	9.5	5.3	4.7	4.3	11.1	4.5	5.2	7.7
REF_0.4	5.7	3.5	4.0	2.5	9.1	na	2.8	7.1
REF_0.5	7.1	18.3	3.4	5.9	7.2	2.8	1.5	6.7
NZ7.5_0.3	7.1	7.0	6.1	24.6	5.4	7.4	4.9	13.4
NZ7.5_0.4	3.7	8.1	2.6	3.8	3.3	3.0	4.7	8.0
NZ7.5_0.5	3.0	0.0	4.4	4.5	3.8	7.9	3.6	8.6
NZ15.0_0.3	7.7	11.7	3.0	16.7	5.1	9.8	3.0	6.0
NZ15.0_0.4	5.1	2.0	4.5	4.4	3.7	5.4	3.5	11.1
NZ15.0_0.5	2.9	3.4	3.8	18.4	2.8	7.1	2.7	8.7
SF7.5_0.3	10.6	5.7	1.9	15.8	6.5	7.7	4.4	7.6
SF7.5_0.4	6.7	11.9	1.4	0.0	3.3	6.1	3.3	13.5
SF7.5_0.5	4.5	8.9	1.8	22.2	4.6	13.0	4.6	5.5
SF15.0_0.3	8.6	5.8	4.2	9.5	8.0	6.3	2.9	41.4
SF15.0_0.4	4.9	2.0	2.2	22.7	5.3	1.9	1.6	37.5
SF15.0_0.5	3.0	6.7	3.7	13.5	5.1	13.7	1.7	11.8

**Table 6 materials-14-01448-t006:** Mass loss measured by thermal analysis (TA) in regards to the main hydration products, chemically bounded water, and hydration degree.

Label	Ettringite 105–160 °C (%)		C-S-H 160– 423 °C (%)		Portlandite 423–500 °C (%)		Phases in Range 500–900 °C (%)		w_ch.b._ (%)		Hydration Degree α (%)	
Days of curing	28	180	28	180	28	180	28	180	28	180	28	180
REF_0.3	3.4	3.5	5.7	6.4	3.5	3.7	3.4	3.1	14.0	14.8	58.4	61.6
REF_0.4	3.4	3.9	5.3	7.1	3.0	4.1	8.7	3.3	15.2	16.4	63.5	68.3
REF_0.5	3.3	4.5	6.8	7.2	4.1	4.2	2.8	5.1	15.3	17.9	63.9	74.5
NZ7.5_0.3	3.4	3.7	6.0	6.5	3.2	3.4	4.0	3.2	14.2	14.9	59.1	62.1
NZ7.5_0.4	3.5	4.0	6.3	7.0	3.3	3.7	3.8	3.0	14.7	16.0	61.1	66.5
NZ7.5_0.5	4.1	4.8	6.7	7.4	3.4	3.7	4.5	3.4	15.9	17.2	66.3	71.7
NZ15.0_0.3	3.9	4.3	5.7	6.9	2.4	2.9	6.9	2.8	14.9	15.3	61.9	63.7
NZ15.0_0.4	4.3	4.6	6.5	6.9	2.8	2.9	4.4	3.3	15.3	15.7	63.8	65.6
NZ15.0_0.5	4.6	5.4	6.2	7.3	2.6	2.8	7.2	5.2	16.4	17.7	68.5	73.6
SF7.5_0.3	3.1	3.5	5.5	6.5	2.8	3.3	5.3	3.2	13.6	14.7	56.7	61.2
SF7.5_0.4	3.4	4.1	6.6	7.2	3.5	3.8	3.7	3.8	15.0	16.7	62.4	69.6
SF7.5_0.5	4.0	4.4	6.3	7.6	3.3	3.9	6.8	4.0	16.4	17.5	68.5	72.9
SF15.0_0.3	3.2	3.7	5.0	5.9	2.2	2.5	5.5	3.3	12.7	13.4	52.8	55.8
SF15.0_0.4	3.8	4.4	5.7	6.9	2.7	3.0	6.8	4.2	14.9	16.0	62.1	66.9
SF15.0_0.5	3.8	4.5	6.2	6.9	2.8	3.0	6.1	6.1	15.2	16.9	63.2	70.4

**Table 7 materials-14-01448-t007:** Assumptions for mineral additive parameters.

	ConsideredAssumption	Real Performance
Type	A	B	C, D
Additive particle	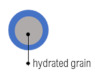	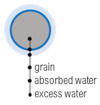	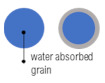	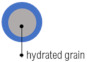
Real binder in the w/b ratio	cem. + add.	cem.	cem.	cem. + add.
Relation to the designed w/b ratio	-	Higher	equal/lower	equal
Participation in hydration	Yes	no	no	yes

## Data Availability

The data presented in this study are available on request from the corresponding author.

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
