# Peer review of "Concept of Evaluation of Mineral Additives’ Effect on Cement Pastes’ Durability and Environmental Suitability"

_materials, 2021, doi:10.3390/ma14061448_

Round 1

Reviewer 1 Report

Manuscript ID:  materials-1133873

Title: Concept of evaluation of mineral additives´ effect on cement pastes durability and environmental suitability  

Comments are marked in the text - the file of the paper

Author Response

Dear reviewer,

we really appreciate your thorough revision. We hope that following revised version of our manuscript will be sufficient.

Authors

Reviewer 2 Report

The manuscript titled “Concept of evaluation of mineral additives´ effect on cement pastes durability and environmental suitability” aims through workability, density, flexural strength, compressive strength, rapid chloride penetration test, water absorption, and thermal analysis to provide an evaluation of mineral additives and their effect in the cement composites regarding the durability and environmental suitability. The study is extensive and provides many various results. From my analysis I found several issues that I’d like to address:

  • Lines 10-11: The first sentence is not easy to read and since Abstract is one of the most important parts of any article, I’d recommend modifying it.
  • Lines 18-19: Based on my own experience, this is actually a very well-known fact.
  • From reading the abstract, I cannot say what is the novelty/originality of the manuscript.
  • “a draft of comprehensive analysis” doesn’t sound scientific.
  • Lines 22-23: I believe that it was meant suitable instead of sustainable based on the title.
  • Lines 27-28: Twice “understood” in one sentence.
  • In my opinion, the introduction is very general and provides mainly the very well-known general knowledge related to the topic. For example, lines 45-59 only summarize the main factors influencing most of the properties, but it is not explained how.
  • Line 58: I am not sure if I understood well what is meant by the high temperature of input materials. Their production temperature or somehow preheating them before mixing them together?
  • Lines 60-64: this should be supported by references.
  • General comment: In my opinion, there are many very long sentences which are difficult to follow.
  • I think that the term “inert component” is too strong in this context. What about material of low reactivity or something similar?
  • Line 74: Low porosity refers to total open porosity or what kind of porosity? I believe that it is important to state this clearly in reaction to the Figure 1b.
  • Lines 74-84: I’d propose separating the points (separated sentences without connecting comma).
  • Lines 91-92: I am not sure if I understand well what is meant.
  • Lines 94-102: The list of properties is very general and covers most of the studied properties in literature. Based on what these properties were selected (references)?
  • Lines 107-109: This statement should be supported by the references.
  • Lines 111-140: The organization of the text here is not clear. I’d recommend summarizing the literature more continuously.
  • Lines 152-155: The novelty of the paper ALSO lies – until now, the paragraph summarizing the manuscript was describing routine measurements of SCMs which have been commonly used in cement research for decades.
  • Lines 154-155: By choosing silica fume for the design of cement pastes, I’d not refer to environmental suitability. The SF is expensive and produced at high temperatures.
  • Line 172: The EN 1008 should be cited (reference).
  • Lines 172-175: To adjusting some of pastes mixtures – It should be more accurate.
  • Based on what the supplement ratios were chosen? 7.5% and 15% seem high for SF.
  • Table 2: The labeling of the samples is very confusing. I’d suggest using more intuitive labeling such as REF = reference, NZ7.5 = pastes with 7.5% of natural zeolite, etc.
  • Line 186: How the materials were mixed? In a mixer of by hand? And which of the pastes were adjusted by the superplasticizer?
  • Line 198: How the oven-dried state is defined? For how long the samples were dried and at what temperature?
  • Line 202: The dimensions are provided in an unusual way, I’d suggest keeping a simple 40x40x160 mm.
  • The Equation 1: Since the compressive strength uses the saturated state as ssd, it might be less confusing, if the mass of saturated sample was labeled as mssd according to this.
  • Line 258: In my own experience, I am convinced that limiting thermal analysis for 800 C in terms of quantifying the decomposition of carbonates is low. Why the graphs are not displayed?
  • Lines 266-270 + 270-272 these statements should be supported with references.
  • Lines 280-284: References are missing.
  • Line 296: I am not sure to understand what “additional plasticization” means.
  • Lines 298-304: Supporting references are missing. The structure of zeolite crystalline grid should be explained more in detail.
  • Lines 310-313: Too long sentence with twice as well.
  • 4-6 are not easy to follow. I’d strongly suggest modifying them into bar charts and merging some of them together. The newly gained space could be used to display thermal analysis results for example. Moreover, the obtained results should be compared to the literature.
  • Lines 323-325: The statement should be supported with references.
  • The equations 4-7 were proposed by the authors? The emphasis should be put on explaining more deeper the main purpose of them and why it is beneficial to use them for the recalculation of the mix composition. I am not sure to understand the difference between w/c and w/b here and the need for their re-computations.
  • Lines 357-367 + Figure 7: How the chemically bound water was computed? In my own experience it is very difficult to separate physically bound water and chemically bound water in the case of cementitious pastes. References helping to understand this part are missing.
  • Lines 387-388: How many results were rejected?
  • Flexural strength: I am not sure to understand why some of the tests were performed on the dried samples. This is very unusual and not practical for any comparison. The obtained results should be discussed in comparison with literature.
  • Figures 8 and 9 are a bit fuzzy (I believe thanks to the shading effect – I’d let it clear without it).
  • The compressive strength results are displayed in Figures 10-12 (the same suggestion as in 34) and the related text only summarizes what can be seen. I’d prefer a discussion/comparison of the results more in detail with literature.
  • Line 430: The reference is missing.
  • Lines 442-443: Comparison to the literature is missing. Moreover, the natural zeolite has been used as SCM for a while.
  • Lines 452-454: I am not sure if I understood well this part.
  • Lines 456-457: It might be beneficial to compare the results to the existing literature to support this statement.
  • Figure 13: The obtained results are limited to the area between 0.3 and almost 0.6. I’d not predict the trend out of this range as it is only assumption.
  • Lines 465-467: The reference is missing.
  • Lines 480 – 482 + 482-484: References are missing.
  • Lines 491-492: References are missing.
  • Line 496: It doesn’t sound scientific. I’d suggest a more detailed insight.
  • Line 528: This “obvious dependence” has been confirmed also by other authors (references are missing).
  • Line 530: Instead of “prediction” I’d rather use “assumption”.
  • Equation 9: Again, it is not clear if the equation is proposed by the authors or taken from the literature. If it is proposed by the authors, the emphasis should be put on explaining it more in detail.
  • Lines 554-557: Unfortunately, the used literature is from 1985 and I currently don’t have an access to it to check it.
  • Lines 568-569 about C-S-H gels are not accurate since in the same temperature interval, the decomposition of ettringite also takes place and the physically bound water is also released. But this cannot be confirmed as the authors did not use the most widely used method for determination of mineralogical composition – XRD.
  • Lines 575-578: References are missing.
  • Table 6. In my opinion, the hydration degrees are very low after 28 and 180 days. I’d suggest to compute the amount of portlandite and carbonate based on the known mass losses and their molar masses.
  • Line 592: Not that much as would expected. Why?
  • Lines 626-629: What does it mean “within one of chemical analysis”?
  • In the section 3.8 there is lack of any literature support.
  • The self-citations are within the reasonable range (6/45).

Author Response

Dear reviewer,

we really appreciate your thorough revision. We hope that following revised version of our manuscript will be sufficient.

Authors.

Reviewer 3 Report

The paper presents an interesting work related to the application of a holistic approach for the evaluation of the addition of mineral additives that are potential supplementary cementitious materials in order to analyse their effects on the durability and sustainability of cement composites. In general, the authors present a well-written and well-structured paper.

Some minor comments:

Introduction

This section presents 43 citations, however, only a few articles are within the last 5 years, not being recommended for an actual state of the art. Some suggestions:

- BERENGUER, R. et al. Durability of Concrete Structures with Sugar Cane Bagasse Ash. Advances in Materials Science and Engineering, v. 2020, 2020.

- Juhart, J., David, G. A., Saade, M. R. M., Baldermann, C., Passer, A., & Mittermayr, F. (2019). Functional and environmental performance optimization of Portland cement-based materials by combined mineral fillers. Cement and Concrete Research, 122, 157-178.

- Pan, G., Li, P., Chen, L., & Liu, G. (2019). A study of the effect of rheological properties of fresh concrete on shotcrete-rebound based on different additive components. Construction and Building Materials, 224, 1069-1080.

Materials and methods

- Table 2 should be better explained

- In items 2.3, 2.5, 2.6 and 2.7, a methodology was performed. Is this methodology standardized? Why wasn’t referred? If it is not standardized, why haven’t you used a standardized methodology? Since there are so many standards for the methodologies used.

- In item 2.4, line 199, why did you use 3 samples for flexural strength and 6 samples for compressive strength? Do you have any statistical explanation? Or some reference?

Results and Discussion

- In table 3, why did the results of mixtures P01, P02 and P03 increase after 60 min?

- Table 3 has some inconsistent data Please create a paragraph with some explanations of the unexpected results.

- the word "immediately" used in line 277, should be replaced by “5 min after mixing”

- In line 280 – 304 of discussion it is necessary to make a few comparisons with results found in the literature.

- Figures 4, 5 and 6 need to have the same magnitudes of x and y axes for better visualization and comparison.

-In relation to Figures 4, 5 and 6, the densities of the D28 mixtures are higher than after the healing D180. What I understand was for densities after 180 days to be lower than compared to after 28 days. Could you explain this phenomenon? The explanation presented between lines 319 – 325 does not seem to convince, needs to bring what some authors in the literature talk about it to enrich the discussion.

- The explanation presented in lines 309 – 313 does not seem correct when it was observed the mixtures P3, P6 and P9 and also check the value of P15 (if it can have outliers in relation to the mean). Continuing to observe the values of P15, it is very important to treat these data, because even in Figure 7 the slope from P13 to P14 is different from P14 to P15.

- Item 3 is very well written when it comes to presenting the results of the research itself, but it needs to bring the DISCUSSION of the results found comparing with the literature so that the statements brought are more con-winning. Therefore review the discussions of items 3.1 to 3.8.

Conclusions

- On Lines 719-720 “Flexural strength should not be regarded as cement paste durability parameter of 719 notice due to its brittleness what leads to low precision of measurements...” This sentence should be completed in a different way. Is it true for all kinds of concrete mixture?

In conclusion, I believe the work is suitable for publication in Materials journals with some minor revisions.

Author Response

(The authors gave the same response as above.)

Round 2

Reviewer 1 Report

The introduced additional explanations and corrections significantly improved the quality of the paper. Now the authors can be satisfied with the result of their work.

Reviewer 2 Report

I would like to thank to the Authors for an extensive effort that has been made towards improving this manuscript. I don't have additional comments or suggestions.